

# Empirical analysis and modeling of Argos Doppler location errors in Romania

Laurentiu Rozylowicz[1], Florian P. Bodescu[2], Cristiana M. Ciocanea[1], Athanasios A. Gavrilidis[2], Steluta Manolache[1], Marius L. Matache[1], Iulia V. Miu[1], Ionut C. Moale[2], Andreea Nita[1] and Viorel D. Popescu[1,3]

[1] Center for Environmental Research and Impact Studies, University of Bucharest, Bucharest, Romania
[2] Multidimension R&D, Bucharest, Romania
[3] Department of Biological Sciences, Ohio University, Athens, OH, USA

Corresponding author
Laurentiu Rozylowicz,
laurentiu.rozylowicz@g.unibuc.ro

## ABSTRACT

**Background:** Advances in wildlife tracking technology have allowed researchers to understand the spatial ecology of many terrestrial and aquatic animal species. Argos Doppler is a technology that is widely used for wildlife tracking owing to the small size and low weight of the Argos transmitters. This allows them to be fitted to small-bodied species. The longer lifespan of the Argos units in comparison to units outfitted with miniaturized global positioning system (GPS) technology has also recommended their use. In practice, large Argos location errors often occur due to communication conditions such as transmitter settings, local environment, and the behavior of the tracked individual.

**Methods:** Considering the geographic specificity of errors and the lack of benchmark studies in Eastern Europe, the research objectives were: (1) to evaluate the accuracy of Argos Doppler technology under various environmental conditions in Romania, (2) to investigate the effectiveness of straightforward destructive filters for improving Argos Doppler data quality, and (3) to provide guidelines for processing Argos Doppler wildlife monitoring data. The errors associated with Argos locations in four geographic locations in Romania were assessed during static, low-speed and high-speed tests. The effectiveness of the Douglas Argos distance angle filter algorithm was then evaluated to ascertain its effect on the minimization of localization errors.

**Results:** Argos locations received in the tests had larger associated horizontal errors than those indicated by the operator of the Argos system, including under ideal reception conditions. Positional errors were similar to those obtained in other studies outside of Europe. The errors were anisotropic, with larger longitudinal errors for the vast majority of the data. Errors were mostly related to speed of the Argos transmitter at the time of reception, but other factors such as topographical conditions and orientation of antenna at the time of the transmission also contributed to receiving low-quality data. The Douglas Argos filter successfully excluded the largest errors while retaining a large amount of data when the threshold was set to the local scale (two km).

**Discussion:** Filter selection requires knowledge about the movement patterns and behavior of the species of interest, and the parametrization of the selected filter typically requires a trial and error approach. Selecting the proper filter reduces the errors while retaining a large amount of data. However, the post-processed data

typically includes large positional errors; thus, we recommend incorporating Argos error metrics (e.g., error ellipse) or use complex modeling approaches when working with filtered data.

## INTRODUCTION

Advances in wildlife tracking technologies allow researchers to track the movement of many terrestrial and aquatic species (*Thomas, Holland & Minot, 2012*). Movement analysis has evolved from short-term local studies on small numbers of individuals to long-term global studies on hundreds of individuals, allowing researchers to answer more complex questions about animal movement and space use (*Block et al., 2011*; *Sequeira et al., 2018*). These data can be included in statistical models and used to understand movement patterns, population redistribution, habitat use, habitat selection, and conservation needs (*Bridge et al., 2011*; *Doherty et al., 2017*; *Hooten et al., 2017*; *Pendoley et al., 2014*; *Pop et al., 2018*; *Schofield et al., 2013*).

Collecting good quality movement data remains a challenging task mainly due to technological constraints (*Hooten et al., 2017*). Well-known tracking technologies such as radio telemetry (VHF telemetry), satellite-based telemetry (GPS, Argos), and light-level geolocation have certain limitations (*Bridge et al., 2011*). The main challenges are the physical size and the mass of the devices. In particular, the mass of the device must not exceed 5% of the animal's body-weight (*Silvy, 2012*). Furthermore, transmitters must be protected from environmental hazards and damage and must include a long-lasting battery or alternative power source for consistent one-way or two-way communication (*Bridge et al., 2011*). As such, devices meeting these parameters can be cumbersome and heavy (*Silvy, 2012*), and not well suited for many small-bodied animals.

The most accurate available tracking technology is the global positioning system (GPS), capable of a horizontal location accuracy of under 10 m (*Madry, 2015*). The weight of present-day GPS receivers varies between a minimum of four g (lifespan limited to a few transmission days and suitable for individuals weighing ~80 g) to in excess of one kg (typically a lifespan averaging 2 years and suitable only for large animals). An alternate option for long-term animal movement studies is the Argos satellite Doppler-based system, which relies on transmitters with extended lifespans that now weigh less than four g, and are able to deliver an unlimited number of locations in near real-time (*Bridge et al., 2011*; *Hooten et al., 2017*; *Thomas, Holland & Minot, 2012*). However, the small size comes at a cost in terms of accuracy of localization compared to GPS. Thus, data interpretation may pose a challenge for inexperienced users (*Rozylowicz et al., 2018*). Regardless of the device size, Argos transmitters, or Platform Transmitter Terminals (PTT), provide locations with the same error rate. In addition, the data requires the application of complex control processes, such as filtering and modeling

(*Thomas, Holland & Minot, 2012*). If the PTT's are equipped with GPS receivers, the location precision can be increased by retaining only validated locations (*Lopez et al., 2015*). Adding a GPS unit to a PTT device results in increasing the minimum device weight to approximately 20 g per unit.

The Argos system utilizes positioning instruments on board six polar-orbiting satellites, in addition to ground-based receiving stations and data processing centers. It delivers near real-time localization of transmitters to end-users. Argos locations are calculated from the Doppler shift of a PTT radio frequency during a satellite pass (*CLS, 2016*). Collecte Localisation Satellites (CLS), the operator of the Argos system, provides several metrics for data quality, such as a location class (LC) based on the number of messages received for each location. The estimated upper bound errors are 250 m for LC 3 (best accuracy class), 500 m for LC 2, 1,500 m for LC 1, and over 1,500 m for LC 0. For locations derived from three or fewer messages (LC A, LC B), Argos does not provide error thresholds. Invalid locations are labeled as LC Z and GPS locations as LC G (*CLS, 2016*). CLS pre-processes these locations using one of Argos's nominal filters such as the Least squares algorithm or the Kalman filter (*Lopez et al., 2015*). In practice, location errors of 10–100 km often occur due to communication conditions driven by the environment or animal behavior (e.g., animal speed, terrain fragmentation, rain, cloud cover, temperature) (*Christin, St-Laurent & Berteaux, 2015*; *Costa et al., 2010*; *Douglas et al., 2012*; *Dubinin, Lushchekina & Radeloff, 2010*; *Sauder, Rachlow & Wiest, 2012*; *Witt et al., 2010*). Thus, filtering the data to exclude implausible Argos locations before employing movement analysis has become a standard approach for researchers (*Hooten et al., 2017*). Furthermore, the quality of data seems to be highly dependent on geographic location. The Argos transmission systems in Eastern Europe are lower in power, their signals being hidden by a background radio noise present across the Argos frequency (*Gros, Malardé & Woodward, 2006*).

Location errors can be filtered using destructive (i.e., removing implausible locations) and reconstructive filters (i.e., evaluation of uncertainty in the estimation of locations) (*Douglas et al., 2012*). Destructive filters remove duplicates (e.g., identical timestamps), locations outside of a defined range (e.g., thresholds for geometric dilution of precision, latitude, longitude, LC), or locations exceeding a fixed movement rate or a turning angle (*Douglas et al., 2012*; *Kranstauber et al., 2011*). One such advanced destructive filter is the Douglas Argos filter algorithm, available on the Movebank database of animal tracking data (*Kranstauber et al., 2011*). The Douglas Argos filter algorithm uses thresholds to mark the outliers as implausible locations. It is available in three settings: the maximum redundant distance filter (MRD) (which retains near-consecutive locations within a distance threshold), the Douglas Argos distance angle filter (DAR) (which retains near-consecutive locations within a distance threshold and location-passing movement-rate and turning-angle tests), and the hybrid filter (HYB) (which combines MRD and DAR parameters specifically for migratory species) (*Douglas et al., 2012*). In contrast, reconstructive filters employ advanced statistical methods to detect animal movement characteristics without removing locations (e.g., discrete-time movement model, correlated random walk state-space models, movement-based kernel density

estimates, Bayesian State-Space Models, and hidden Markov models), or model the data using the errors associated with movement (e.g., Argos error ellipse) (*Hooten et al., 2017*; *Jonsen, Flemming & Myers, 2005*; *Lopez et al., 2015*; *Silva et al., 2014*).

Considering the behavioral, environmental, and geographic specificity of the errors associated with Argos data and the lack of benchmark studies in Eastern Europe, the research objectives for this study are: (1) to provide empirical evidence of the accuracy of locations collected via the Argos Doppler system in Romania, (2) to investigate the effectiveness of straightforward destructive filters for improving Argos data quality, and (3) to provides guidelines for processing Argos wildlife monitoring data in Eastern Europe. The errors associated with Argos locations were assessed in four geographic locations from Romania under three spatial movement conditions: static, low-speed and high-speed. The effectiveness of the Douglas Argos distance angle filter algorithm was then evaluated in terms of location error minimization.

## MATERIALS AND METHODS

### Trial sites

The accuracy of Argos locations was analyzed in four areas in Romania. These areas varied both topographically and in terms of reception conditions and could be categorized as:

(1) Urban: two urban parks within a residential area of Romania's largest metropolis, Bucharest (within the Tineretului Park for the static test, 44°24′N 26°06′E; and the shoreline of Vacaresti Lake for the mobile tests, 44°24′N, 26°07′E).

(2) Unobstructed flat rural lowland: Saveni, Ialomita County (44°35′, 27°37′E).

(3) Highly fragmented rural topography: Iron Gates Natural Park, Mehedinti County (44°41′N 22°21′E), along the Danube River.

(4) Moderately fragmented rural upland: Sighisoara (within Breite for the static tests, 46°12′N 24°45′E; and Sighisoara-Apold for the mobile tests, 46°09′N 24°46′E)).

At all trial sites, three tests were carried out: a static test, a low-speed test, and a high-speed test (Fig. 1).

### Experimental protocol

Five Argos PTTs were used, all identical GeoTrack 23g Solar PTT units (GeoTrack Ink., Apex, NC, USA) with a repetition period of 60 s. The PTTs were initially programmed for an 8 h "On" and 43 h "Off" transmission cycle. They were then manually activated at the start of the working day and restarted if the transmission cycle lasted longer than 8 h. Daily activation occurred 10 min before the first Argos satellite was scheduled to be visible, according to the satellite pass prediction (*CLS, 2016*). During the experiment, a transmission day started after the activation of the PTTs, when the first Argos message was received by a given PTT, and ended after a minimum of 6 h, when the satellite transmitting the last message was no longer visible to the PTTs. The PTTs were glued on a stake 20 cm from each other, with their antennas pointing in the same direction toward the sky. Argos messages were processed by CLS using the standard

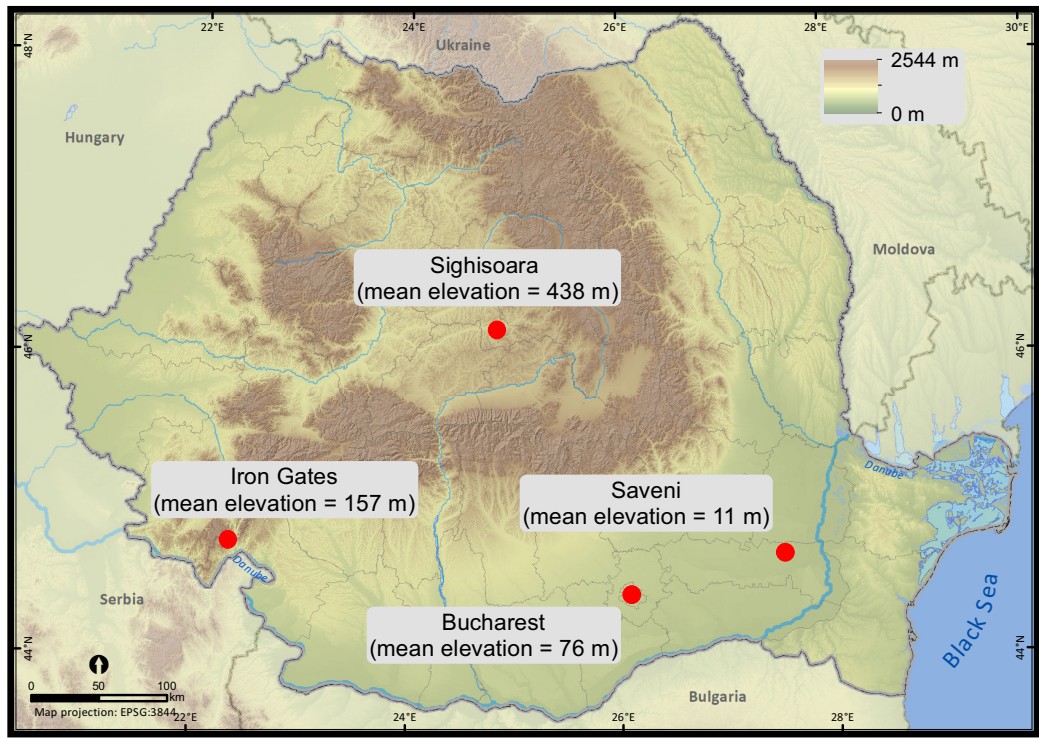

**Figure 1 Trial sites for the motion-controlled tests within Romania (static, low-speed, high-speed).**

Kalman filter algorithm (*CLS, 2016*), with a predefined average speed of 16 m/s. To estimate the accuracy of the Argos locations, each Doppler location was matched with a GPS location (Garmin Oregon 650; Garmin Ltd., Olathe, KS, USA) obtained within a 5-min time period. Garmin GPS receivers have a precision of under 10 m in low-rise residential areas (*Beekhuizen et al., 2013*). GPS locations could thus be considered accurate.

Each speed test lasted a total of 6 transmission days, with a minimum of 6 transmission hours per cycle. For static tests, the five PTTs were positioned 30 cm above ground in unobstructed transmission conditions. For low-speed tests, researchers carried the PTTs attached to a backpack at normal walking speed (four to five km/h). For the high-speed tests, the PTTs were mounted on a bicycle traveling at 15 km/h. In each test, the GPS receiver was set to record a location every 30 s.

## Data processing and analyses

Argos messages were downloaded daily. Each Argos message was assigned to the corresponding trial site and movement test. Prior to statistical analysis, the dataset was cleaned to eliminate records without coordinates or with identical timestamps.

The magnitude of spatial errors was estimated using several error metrics. Distances between Argos locations and the corresponding GPS locations were calculated as the geodesic distance using the WGS 1984 reference ellipsoid (i.e., *location error*, in meters). The direction of the error was calculated as bearing along a rhumb line between the Argos and GPS locations (i.e., *error bearing*, 0–360°) (*Hijmans, Williams & Vennes, 2017*).

Latitudinal and longitudinal errors were calculated as the difference between the UTM coordinates of Argos latitude and longitude and the corresponding GPS latitude and longitude (i.e., *Latitudinal* and *Longitudinal errors* in km). Furthermore, Argos locations were classified as *"in"* or *"out"* of Argos ellipse error by plotting them in the ArcGIS 10.3 (ESRI, Redlands, CA, USA) software package along with the ellipse error components provided by CLS for each location (Data S1).

The variability of log-transformed location errors was evaluated using linear-mixed effects models with *motion* (speed: static, low-speed, high-speed), *place* (trial sites: Saveni, Bucharest, Sighisoara, Iron Gates), and a *terrain ruggedness index* (*Riley, DeGloria & Elliot, 1999*) as fixed effects and the *receiving points* (locations generated simultaneously by all transmitting PTTs at a satellite pass) nested within the *satellite* detecting the PTTs locations as random effects. Grouping the locations by receiving points allowed control of pseudo replication and latent variation unaccounted for by the fixed effects (*Harrison et al., 2018*).

Several linear-mixed effects models were fitted using different combinations of fixed effects with the same nested random effects structure using function *lmer* with restricted maximum likelihood in the package *lme4* (*Bates et al., 2015*) in program R (*R Development Core Team, 2018*). To select the best model predicting the variance of log-transformed location errors we used Akaike's information criterion, corrected for small sample size (Table S1). The model fit was evaluated as the variance in the data explained by fixed effects (marginal $R$-squared) and collectively by fixed and random effects (conditional $R$-squared) (*Nakagawa & Schielzeth, 2013*).

To evaluate the effectiveness of the data filters to minimize the location errors when the tracked species move within a given site, we partitioned the data by trial sites and then ran distance, angle, and rate DAR on Movebank tracking platform (www.movebank.org) (*Douglas et al., 2012*; *Kranstauber et al., 2011*). The two other versions of the Douglas Argos filter are either a truncated version of the distance, angle and rate filter (MRD filter) or specifically designed for migration movement (HYB filter), which was not the case in this experiment (*Douglas et al., 2012*).

The DAR filter was applied twice: with the threshold radius set to encompass two locations which were considered self-confirming (MAXREDUN) at two km (DAR 2) and at 15 km (DAR 15). Smaller threshold values, such as two km are suitable for local scale movement while larger values (e.g., 15 km) are suitable for regional scale (*Douglas et al., 2012*). It is recommended by the authors of the Douglas Argos algorithm to retain locations above LC 1 (locations with higher accuracy class); however, because our test was focused on the effectiveness of this filter at eliminating location errors from the whole dataset, a LC threshold was not provided. The two other user-defined parameters of the DAR filter were kept fixed in both filtering sessions because the experimental conditions were similar. MINRATE (maximum sustained rate of movement over a period of several hours) was set at 15 km/h (maximum velocity achieved during the test) and RATECOEF was set at 10 (specifically used for movements following a very circuitous pattern). MINRATE confirms the plausibility of a location based on a reasonable rate of movement that the animal of interest might sustain and RATECOEF is a scaling parameter which
**Table 1** Location error metrics for all Argos location classes (3,705 valid locations received on four trial sites within Romania, during three motion-controlled tests).

| Location class | Sample size | Mean error (stdev), meters | 68th percentile of errors, meters | Mean error longitude (stdev), meters | Mean error latitude (stdev), meters | % locations in error ellipse | % locations out of error ellipse |
|---|---|---|---|---|---|---|---|
| LC 3 | 528 | 578.61 (802.52) | 520.85 | 466.71 (744.47) | 254.79 (375.95) | 10.42 | 89.58 |
| LC 2 | 520 | 1,230.64 (1,281.56) | 1,383.81 | 969.24 (1,099.82) | 580.09 (818.49) | 4.62 | 95.38 |
| LC 1 | 674 | 2,222.78 (2,466.15) | 2,280.64 | 1,784.74 (2,158.28) | 1,010.66 (1,467.00) | 5.64 | 94.36 |
| LC 0 | 376 | 7,127.17 (14,869.71) | 5,877.38 | 6,195.25 (14,492.92) | 2,630.38 (4,064.45) | 11.17 | 88.23 |
| LC A | 505 | 3,669.57 (6,816.14) | 2,981.35 | 2,894.15 (6,484.07) | 1,622.35 (2,617.89) | 9.90 | 90.10 |
| LC B | 1,102 | 5,717.70 (10,456.50) | 4,820.25 | 4,444.84 (9,611.44) | 2,739.37 (4,725.23) | 28.04 | 71.96 |
| Total | 3,705 | 3,583.66 (8,225.96) | 2,758.73 | 2,872.49 (7,677.60) | 1,604.44 (3,272.32) | 13.98 | 86.02 |

influences how the angle between two consecutive locations is evaluated (parameter values vary between 10, for species displaying out-and-back movement, and 40, for species with directional movement) (*Douglas et al., 2012*). The results of the two user-specific filters (DAR 2 and DAR 15) were compared to the unfiltered data. For comparison purposes, the results for two intermediary MAXREDUN thresholds are presented in a supplementary file: five km (DAR 5) and 10 km (DAR 10).

We used R 3.5.1 (*R Development Core Team, 2018*) package *dplyr* (*Wickham et al., 2018*) for cleaning data, *geosphere* (*Hijmans, Williams & Vennes, 2017*) for calculating error metrics, *dunn.test* (*Dinno, 2017*) for testing differences between various datasets, *lme4* (*Bates et al., 2015*), *MuMIn* (*Barton, 2018*), *merTools* (*Knowles, Frederick & Whitworth, 2018*) for linear-mixed effects models, *ggpubr* (*Kassambara, 2018*), *ggeffects* (*Lüdecke, 2018*), and *openair* (*Carslaw & Ropkins, 2012*) for figures.

## RESULTS

Between June 2017 and September 2017, the five PTTs received 3,705 valid Argos locations (Data S1). Each PTT generated a similar number of locations (min = 717, max = 760, $\chi^2$ (d$f$ = 4, $n$ = 3,705) = 1.86, $p$ = 0.76). For each location, the Argos PTTs transmitted between 1 and 14 messages to one of the six polar-orbiting satellites fitted with Argos instruments. The Argos satellites resulted in the calculation of a dissimilar number of locations (min = 310, NOAA-N, NP′; max = 890, NOAA-18, NN, $\chi^2$ (d$f$ = 5, $n$ = 3,705) = 399.87, $p$ < 0.001).

The dataset was dominated by low-quality data, with over 29% of locations labeled as LC B. 46% of the locations were classified by the CLS as error bounded (Argos LC 3, 2, and 1), from which 14.25% were of high estimated quality (LC 3, <250 m estimated accuracy) (Table 1).

The empirical mean location error for the five PTTs was 3,583.66 m (stdev = 8,225.96 m). Location errors differed significantly by Argos LCs (Kruskal–Wallis chi-squared = 1,170.95, d$f$ = 5, $p$ < 0.001), except for the LC 1 and LC A which showed identical ranking. All the error-bounded LCs (Argos LC 3, 2, and 1) had measured errors which were significantly larger than the location-class-specific 68th percentile estimated by CLS.

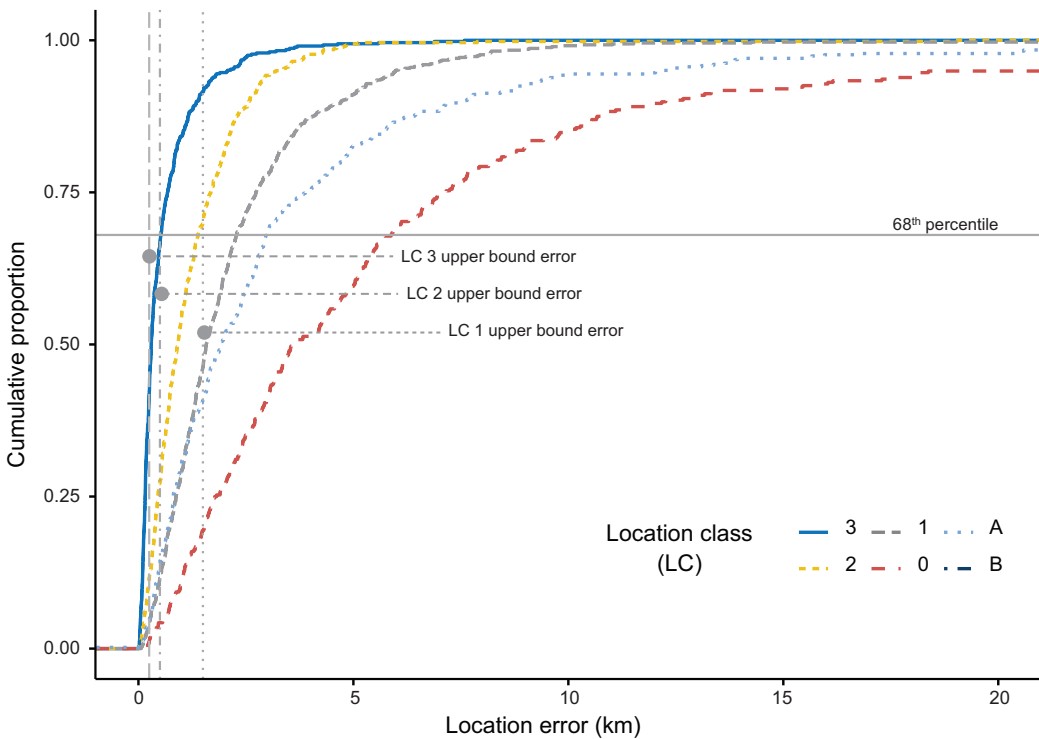

**Figure 2 Cumulative distribution of Argos location errors (km) partitioned by Argos location classes (LC).** The 68th percentile of measured errors is larger than the 68th percentile provided by Argos CLS for error bounded LCs (upper error LC 3, LC 2, and LC1).

Argos LC A and LC B had smaller associated errors than LC 0, a LC which is considered more accurate by CLS (Fig. 2; Table 1).

On average, longitudinal errors were larger (mean = 2,872 m, stdev = 7,678 m) than latitudinal errors (mean = 1,604.4 m, stdev = 3,272.3 m). This pattern was found to be consistent in all LCs (Table 1). The largest proportion of longitudinal errors were in LC 0 (73.96% of locations) and LC 1 (73.44% of locations), while the largest proportion of latitudinal errors was found in the LC B (39.29%) and LC A (38.42%) classes. Geographically, most errors were to the East and West, followed by the North-East and South-West. Those oriented toward the North and South were less present in the dataset (Figs. S1 and S2).

The best linear mixed model showed that the fixed factors *Motion* (speed tests) and *Place* (trial areas) had a significant impact on the errors (Table S1). The fixed factors Motion and Place accounted for 17.45% of the variance in the error data (marginal $R$-squared), while the fixed and random structures (*reception points* nested in the *satellite*) combined accounted for 56.79% of the variance (conditional $R$-squared). The proportion of variance in errors accounted for by the reception points nested in the satellite (Intra Class Correlation Coefficient) was 41.91%, while the satellites themselves accounted for only 5.73%, suggesting that reception conditions at transmission time strongly influenced the quality of the data (Table 2).

A comparison of the confidence intervals of the fixed factors showed that the locations from motion-controlled trials differed significantly. Errors from high-speed tests were

**Table 2 Summary of best mixed effect model (log errors ~ Motion−1 + Place + (1|Satellite/Reception point).**

| Parameter | β | SE | t-value | Lower CI | Upper CI |
|---|---|---|---|---|---|
| Static | 2.89 | 0.06 | 48.35 | 2.771 | 3.016 |
| Low-speed | 0.38 | 0.03 | 11.74 | 0.319 | 0.446 |
| High-speed | 0.57 | 0.03 | 17.08 | 0.501 | 0.630 |
| Bucharest | −0.03 | 0.04 | −0.90 | −0.103 | 0.038 |
| Iron Gates | 0.13 | 0.04 | 3.65 | 0.061 | 0.205 |
| Sighisoara | −0.02 | 0.04 | −0.45 | −0.093 | 0.058 |

**Note:**
Saveni trial site is kept as reference.

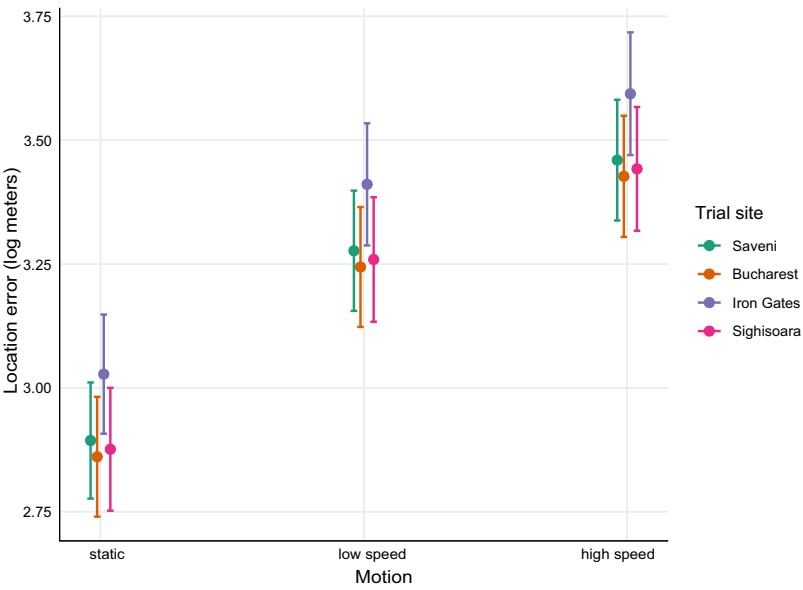

**Figure 3 Mean (±95% CI) fitted values for the best mixed-effects model predicting Argos location errors by *Motion* and *Place* (trial site).**

larger than those in the low-speed tests, and errors in the low-speed tests were larger than in the static tests. The mean error for the static tests was 2,708.84 m, the mean error for the low-speed tests was 3,779.73 m, and the mean error for high-speed tests was 4,550 m (Fig. 3; Table 3). The trial sites also contributed to the error variance, with locations from Iron Gates (a narrow valley) generating larger errors when compared to locations from the other sampling areas. As a comparative example, the mean error for Iron Gates was 4,698.35 m and the mean error for Saveni was 3,122.01 m (Fig. 3; Table 4). Interestingly, it was found that substantial errors existed in the static tests from Iron Gates which were as large as the location errors associated with the low-speed tests conducted in Saveni, Bucharest, and Sighisoara (Fig. 3).

The DAR filter applied to raw data, successfully excluded the largest errors when MAXREDUN was defined for local (two km, DAR 2) and regional (15 km, DAR 15) scales of study. However, the DAR 2 filter was more effective at excluding large errors, by retaining 84.35% of the initial locations compared to 94.82% for the DAR 15 filter

**Table 3 Location error metrics in the three motion-controlled tests carried out within Romania.**

| Motion | Sample size | Mean error (stdev), meters | Mean error longitude (stdev), meters | Mean error latitude (stdev), meters | % locations in error ellipse | % locations out of error ellipse |
|---|---|---|---|---|---|---|
| Static | 1,469 | 2,708.84 (9,588.76) | 2,315.32 (9,215.04) | 1,042.22 (2,810.01) | 16.81 | 83.19 |
| Low-speed | 1,137 | 3,779.73 (7,779.31) | 2,879.02 (6,871.40) | 1,851.93 (3,977.74) | 11.96 | 88.04 |
| High-speed | 1,099 | 4,550.15 (6,381.86) | 3,610.44 (5,958.59) | 2,099.89 (2,909.25) | 12.28 | 87.72 |

**Table 4 Location error metrics in the four trial sites within Romania.**

| Place (trial site) | Sample size | Mean error (stdev), meters | Mean error longitude (stdev), meters | Mean error latitude (stdev), meters | % locations in error ellipse | % locations out of error ellipse |
|---|---|---|---|---|---|---|
| Saveni | 1,106 | 3,122.01 (5,862.75) | 2,489.22 (5,427.42) | 1,410.09 (2,539.10) | 13.29 | 86.71 |
| Bucharest | 969 | 3,311.57 (7,146.49) | 2,595.00 (6,438.32) | 1,545.76 (3,379.37) | 16.10 | 83.90 |
| Sighisoara | 734 | 3,277.75 (6,439.44) | 2,615.05 (5,982.89) | 1,529.57 (2,690.64) | 12.26 | 87.74 |
| Iron Gates | 896 | 4,698.35 (12,113.6) | 3,856.52 (11,495.36) | 1,969.13 (4,229.34) | 13.95 | 86.05 |

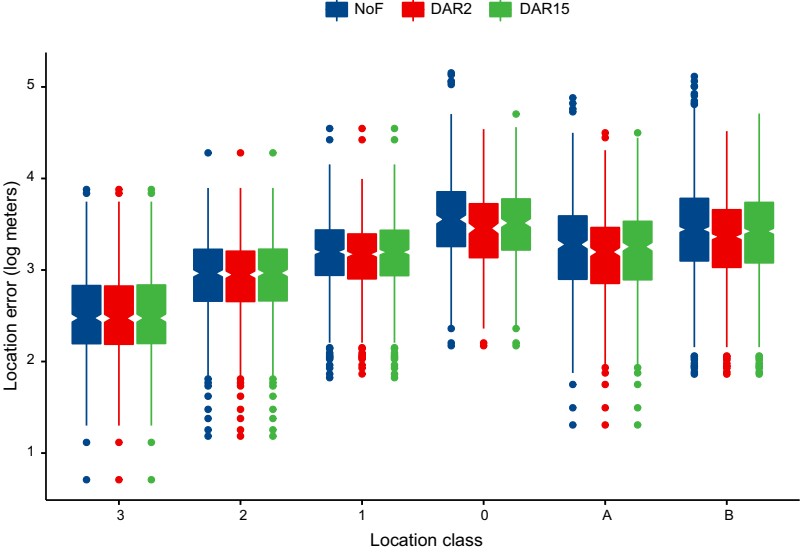

**Figure 4 Effectiveness of Douglas Argos filter (DAR) in moderating Argos location errors by location class.** NoF = unfiltered data, DAR 2 = Douglas Argos DAR with MAXREDUN = two km, DAR 15 = Douglas Argos DAR with MAXREDUN = 15 km.

(Tables S2 and S3). The mean error for the DAR 2 filtered data was 2,313.51 m (stdev = 3,134.67), a 35.45% improvement compared to the error associated with the raw data, while the improvement after the DAR 15 filter was applied was only 27.05%. The DAR 15 filter retained almost all the locations in LC 3, LC 2, and LC 1 classes, while the DAR 2 filter altered the number of LC 2 and LC 1 locations slightly. The most heavily-impacted LC was LC 0—the class with the most substantial errors amongst the data—with only 68.35% of locations retained by the DAR 2 filter and 90.42% by the DAR 15 filter (Tables S2 and S3; Fig. 4). The intermediate thresholds (DAR 5 and DAR 10)

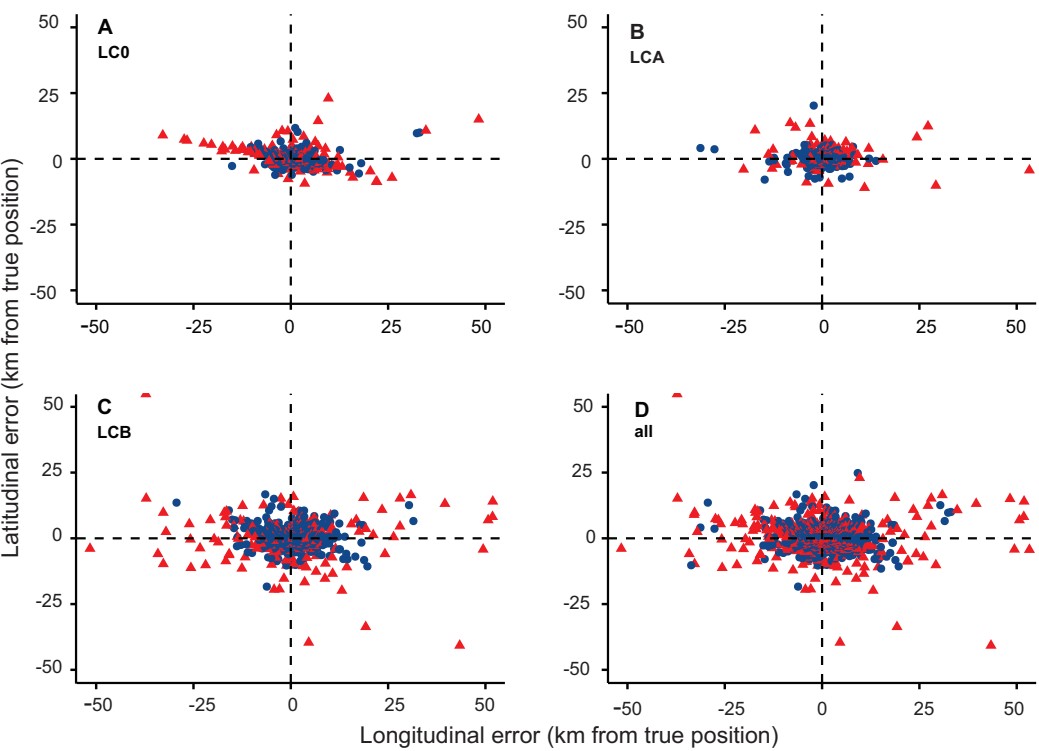

**Figure 5** **Latitudinal and longitudinal errors (km from GPS locations) for (A) LC 0, (B) LC A, (C) LC B, and (D) all LCs (red = rejected Argos locations; blue = accepted Argos locations).**

did not improved the data accuracy in comparison to DAR 2, and the results were only marginally different when compared to DAR 15 (Fig. S3). Douglas Argos algorithms filtered longitudinal and latitudinal errors equally; thus, the filtered data are also anisotropic, with the longitudinal errors larger than the latitudinal errors (Fig. 5).

## DISCUSSION

The accuracy of the Argos Doppler locations received from Romania was negatively influenced by the movement speed and topographic characteristics of the trial sites. The empirical data showed that Argos locations yielded a lower accuracy, even in stationary tests performed in unobstructed areas. This suggests that Argos Doppler telemetry data must undergo a comprehensive filtering process before being used in movement analysis.

In our experiment, 14% of locations were considered category LC 3, the most accurate Argos LC (*CLS, 2016*). However, the 68th percentile of LC 3 locations was twice as large as the 68th percentile provided by CLS as the upper bound error (520.85 vs. 250 m). All error-bounded Argos locations classes (LC 2, LC 1, and LC 0) included larger positional errors than those indicated by *CLS (2016)*, which is in line with the results demonstrated in other controlled and real-life studies. For example, a stationary and mobile test in Southern Russia (*Dubinin, Lushchekina & Radeloff, 2010*) and tests on marine species (*Costa et al., 2010*) yielded errors similar to ours for LC 3 data. Data recovered from these

other studies and our results indicate higher errors than those indicated by CLS for all LCs. LC 0 was the most inaccurate LC (68th percentile = 5,877.38 m), which corroborates the results of other studies (*Douglas et al., 2012*; *Lowther et al., 2015*). This suggests that LC 0 locations must be filtered together with LC A and LC B, and should not be considered an accurate LC. Argos errors are not isotropic, and longitudinal errors were larger than latitudinal errors (*CLS, 2016*; *Douglas et al., 2012*), as already reported by all benchmark studies (*Lowther et al., 2015*; *Sauder, Rachlow & Wiest, 2012*; *Witt et al., 2010*). In this study, the mean latitudinal errors for LC 3, LC 2, and LC 1 were only slightly larger than the CLS 68th percentile for the respective LCs. However, these data are not likely to be useful for movement studies, such as home-range analysis (*Hooten et al., 2017*) since longitudinal errors were significant even in optimal reception environments (e.g., flat areas, unobstructed by vegetation). South-Eastern Europe is considered an area with poor reception quality due to the broadband noise covering the Argos 401.65 MHz ± 30 kHz frequency (*Gros, Malardé & Woodward, 2006*), which might have a negative impact on quantity and quality of data. Since these errors were similar to those obtained in other studies outside of Europe, the broadband noise affecting Southeastern Europe (*Gros, Malardé & Woodward, 2006*) seems to have had only a minimal influence on the accuracy of Argos data.

The accuracy of Argos Doppler locations is influenced by a plethora of factors such as the PTTs' repetition rate, topography, vegetation, terrain ruggedness, electromagnetic noise, and geographic area (*Freitas et al., 2008*; *Lowther et al., 2015*; *Nicholls, Robertson & Murray, 2007*; *Sauder, Rachlow & Wiest, 2012*). The best linear mixed effects model for this study showed that the movement speed of the PTT had the most significant influence on Argos location errors, while the trial area contributed only marginally to the variation in errors. As expected, motion-controlled tests generated significantly varying errors. Static tests generated smaller errors than low-speed tests, and high-speed tests generated larger errors than low-speed tests. However, topographical obstruction of the sky influenced data acquisition. Data obtained from the highly-fragmented Iron Gates trial area contained larger errors than the other three trial areas, including Bucharest city potentially affected by electromagnetic interference (*Gros, Malardé & Woodward, 2006*). In the Iron Gates area, the static test generated positional errors as large as in the low-speed motion tests performed in the other three areas. This suggests that locations from fragmented areas may be highly imprecise and can lead to biased conclusions about animal movement and location if not adequately filtered (*Lopez et al., 2015*). The variance explained by the random part of the linear mixed effects model suggests that the satellite detecting the location of the PTT had a minimal impact on accuracy, while the dominant source of positional errors were probably due to other random factors, such as poor line of sight to the satellite as a result of local topography, the presence of obstructing vegetation or the relative orientation of the respective PTT with respect to the sky (*Christin, St-Laurent & Berteaux, 2015*; *Doherty et al., 2017*; *Dubinin, Lushchekina & Radeloff, 2010*; *Soutullo et al., 2007*).

Due to the large positional errors, Argos Doppler data should be filtered or modeled, considering the uncertainty of locations (*McClintock et al., 2014*). Data filtering is a

challenge, as the aim is to reduce low-quality data as much as possible while retaining the necessary amount of data for analysis (*Hooten et al., 2017*). In the filtering exercise conducted in this study, we tested the effect of the Douglas Argos distance, angle, and rate filter. This filter retains spatially redundant locations, passing movement rates, and turning angle tests (*Douglas et al., 2012*). The results indicated that the selection of a proper self-validating distance threshold significantly reduces the errors while retaining a large amount of data. In this study, a larger threshold, MAXREDUN = 15 km, reduced the efficacy of the filter considerably by retaining 10% more locations than when the threshold was set at two km. The differences between the two approaches suggested that previous knowledge of movement behavior of species of interest is essential to ensure good quality data. For example, if the species of interest is known to perform frequent long-distance movements, then a larger MAXREDUN is required. The DAR filter was tested by targeting all the LCs; however, LC 3, LC 2, and LC 1 were only slightly impacted, and thus it is recommended to run the filter using LC 1 as the threshold LC as suggested by *Douglas et al. (2012)*.

Even if selecting the optimal threshold, the post-processed data may include large positional errors; therefore, we recommend incorporating Argos error metrics such as error ellipse into models (*McClintock et al., 2014*). As an alternative, complex statistical approaches such as state-space modeling can be used instead of standard movement analysis (*Hooten et al., 2017*). While the results were provided based on the distance, angle, and rate filter which fitted the data, other available filtering approaches might be more effective for a given species (e.g., speed filters, Douglas Argos MRD, Douglas Argos HYB).

## CONCLUSIONS

To provide guidance for processing Argos Doppler-derived locations obtained from Eastern Europe, we assessed the errors associated with Argos locations in four geographic locations in Romania using static, low-speed and high-speed tests. The effectiveness of destructive filters was evaluated in terms of the minimization of location errors, using the Douglas Argos distance angle filter algorithm as an example. The results indicated that the received Argos locations had larger positional errors than those indicated by the operator of the Argos system. This included cases where the reception conditions were ideal. The magnitude of the errors varied among LCs; however, locations in the LC 0 class were prone to large errors. Positional errors were anisotropic, predominantly oriented East and West, which resulted in larger longitudinal errors. Errors were mostly related to the speed of the transmitters and the reception conditions at the time of transmission (e.g., the orientation of the transmitter with respect to the sky), but other factors such as topography contributed to receiving low-accuracy data as well. The destructive Douglas Argos distance angle filter removed between 15% (self-confirming distance threshold = two km) and 5% (threshold = 15 km) of locations. However, the mean errors remained larger than those indicated by the operator of Argos system. The Argos data should therefore be used with caution in movement ecology studies, especially for species with small home ranges, such as songbirds, reptiles, or small mammals. Filter selection for data processing requires knowledge about the movement patterns

and behaviors of the species of interest, and parametrization of the selected filter must follow a trial and error approach. Argos Doppler is unsuitable for movement analysis of species with limited movement capabilities. GPS tags would be preferable for such species, enabling detailed analysis of habitat selection and movement, with the caveat that store-on-board systems might be required owing to the larger sizes of the units that send data remotely via satellite.

### Funding

The research was supported by a grant of the Romanian National Authority for Scientific Research (www.uefiscdi.ro), PN-III-P2-2.1-PED-2016-0568—Argos based applications for real-time wildlife monitoring in Romania (BioMoveFix). The funders had no role in study design, data collection and analysis, decision to publish, or preparation of the manuscript.

### Grant Disclosures

The following grant information was disclosed by the authors:
Romanian National Authority for Scientific Research (www.uefiscdi.ro), PN-III-P2-2.1-PED-2016-0568—Argos based applications for real-time wildlife monitoring in Romania (BioMoveFix).

### Competing Interests

Florian P. Bodescu, Athanasios A. Gavrilidis, and Ionut C. Moale are employed by Multidimension SRL, Bucharest.

### Author Contributions

- Laurentiu Rozylowicz conceived and designed the experiments, performed the experiments, analyzed the data, contributed reagents/materials/analysis tools, prepared figures and/or tables, authored or reviewed drafts of the paper, approved the final draft.
- Florian P. Bodescu performed the experiments, contributed reagents/materials/analysis tools, approved the final draft.
- Cristiana M. Ciocanea performed the experiments, approved the final draft.
- Athanasios A. Gavrilidis performed the experiments, analyzed the data, contributed reagents/materials/analysis tools, prepared figures and/or tables, authored or reviewed drafts of the paper, approved the final draft.
- Steluta Manolache performed the experiments, contributed reagents/materials/analysis tools, approved the final draft.
- Marius L. Matache performed the experiments, approved the final draft.
- Iulia V. Miu performed the experiments, contributed reagents/materials/analysis tools, approved the final draft.
- Ionut C. Moale performed the experiments, approved the final draft.
- Andreea Nita conceived and designed the experiments, performed the experiments, analyzed the data, contributed reagents/materials/analysis tools,

prepared figures and/or tables, authored or reviewed drafts of the paper, approved the final draft.

- Viorel D. Popescu analyzed the data, contributed reagents/materials/analysis tools, prepared figures and/or tables, authored or reviewed drafts of the paper, approved the final draft.

## Data Availability
The raw data and results are available in the Supplemental Files.

## Supplemental Information
Supplemental information for this article can be found online at http://dx.doi.org/10.7717/peerj.6362#supplemental-information.

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
