# Peer review of "Empirical analysis and modeling of Argos Doppler location errors in Romania"

_PeerJ, doi:10.7717/peerj.6362_

## Round 0.1 · original submission · Major Revisions

Congratulations on a paper well reviewed. Each of three reviewers agree the manuscript merits publication in PeerJ. I agree with this assessment. However, Reviewer 1 has some important feedback to consider, namely the jump from filters of 2km to 15km without any reported evaluation between. I suggest the authors review that carefully and respond clearly or repeat that analysis with a range of values. Reviewers 1 and 3 also provided important feedback in annotated PDFs and I encourage the authors to revisit those comments.

Overall, the work required to overhaul the science is limited to addressing Reviewer 1. However, all of the reviewers have requested clarification of the language and improved definitions of terms and acronyms. Please work through the English of the narrative while revising.

Reviewer 1 ·

Basic reporting

Grammar errors and small mistakes can be found over all over the manuscript and can sometimes hinder our understanding. Some large omissions and errors can be found and give the feeling the manuscript was rushed.e.g.: line 187, authors defined DAF 2 but forgot to define DAF 15 just after even though they refer to it later.
The authors should then spend more time making sure everything is perfectly defined and well written.

Overall the clarity of the article could be improved by using well defined terminology. For example, the authors refer several times to “transmission days” without ever defining what this means and to this day I still don’t understand what this refers to… The manuscript could also be easier to read and understand if the authors replaced some expressions such as “area of reception” (could be replaced by “site”) or “motion-controlled tests” which is vague and most of the time useless and could just be referred to as “tests” since no other tests were performed.

Background is clear and well documented even though a quick overview of how the Argos system works could be of interest for uneducated readers.

Figures are well made even though Figure 2 could be smoothed a little.
There is redundancy in figures and tables though, and some could be removed easily or move in supplementary material without harm.

The discussion section of the abstract is very vague and short and could be a little more substantial.

Experimental design

Methods should in my opinion be separated in named subsections (e.g: study sites, statistical analysis) because as it is it can be really confusing. Overall, the whole method section could be clearer.
The methods seem quite exhaustive except for the part about mixed models where the variables used were never cited nor described. A quick listing or a table with the different models used is required.
I believe the article would also gain to test different values of MAXREDUN. The authors test the filters for 2 and 15 km but this leaves a huge gap between the two. Different filters could be made to examine the effect of MAXREDUN on data accuracy. Maybe there is an optimal value that offers the best compromise between loss of data and accuracy gain.
Also the authors note that three different filters are offered by Movebank (line 182-183) yet they only use one without justifying why. Maybe a comparison of the type of filters could have been nice, especially since this is one of the objectives of the paper.

Validity of the findings

The results seem solid but clarity could be improved (see previous comments on figures and terminology and the use of subsections).
The conclusion section essentially just repeats what was said in the discussion section and is therefore quite useless as is.

Additional comments

While the science behind the article seem sound and it appears the authors know what they are talking about, the form could greatly be improved for a better understanding of the paper.
The article provides so many figures and tables that it is sometimes hard to make sense of it. This is not help by grammar errors and omissions all over the paper.
I recommend spending more time trying to make the paper more accessible and make sure every variable is well defined.
Additional filters could also be added to the study to improve its added value.

Annotated reviews are not available for download in order to protect the identity of reviewers who chose to remain anonymous.

Reviewer 2 ·

Basic reporting

The English language needs checking by a native English speaker

I have suggested literature that should be cited in the text.

The structure, figures, table were all clear.

It is clear and self contained


Items to amend
Line 51, here cite key papers, e.g.
Block et al. 2012. Tracking apex marine predator movements in a dynamic ocean. Nature volume 475, pages 86–90
Sequeria et al 2018. Convergence of marine megafauna movement patterns in coastal and open oceans. Proceedings of the National Academy of Sciences https://doi.org/10.1073/pnas.1716137115

Line 55, also cite these papers here:
Pendoley et al (2014). Multi-species benefits of a coastal marine turtle migratory corridor connecting Australian MPAs. Marine Biology 161, 1455-1466. DOI 10.1007/s00227-014-2433-7
Schofield et al (2013). Evidence based marine protected area planning for a highly mobile endangered marine vertebrate. Biological Conservation 161, 101-109 doi.org/10.1016/j.biocon.2013.03.004.

Line 130. The wording does not flow here: “were glued on stake 20 cm each other”

Experimental design

All of the items are covered adequately.

Validity of the findings

All of the items are covered adequately.

Additional comments

General
This is a useful paper investigating Argos doppler location errors in Romania. I have made some suggestions on citations that need to be included. I would also advise that the English language is proof read by a native English speaker. Otherwise the structure is good and the tables and figures are presented well. This paper should make a useful contribution to the journal.

·

Basic reporting

See annotated .pdf

Experimental design

See annotated .pdf

Validity of the findings

See annotated .pdf

Additional comments

See annotated .pdf

---

## Round 0.2 · Minor Revisions

Congratulations. You have done a good job when improving your manuscript as suggested by reviewers. All three reviewers are really positive. One reviewer suggested to accept this manuscript as it is, but two other reviewers noticed few minor issues. They recommend to accept this manuscript after minor revisions.

Could you please check these things before the final acceptance. Reviewers said that you need a figure citation in the line 274. In addition, you could add a brief explanation of what transmission day or transmission hour means. You should also check few paragraph presentations, issues about extra space hits, and you have mixed US and UK English styles. You can read all these comments from the review reports provided by our reviewers.

Aftre these minor modifications, upload a final revised version to the manuscript system of PeerJ.

Reviewer 1 ·

Basic reporting

The readability of the article has really been improved through the changes to the text.

Only 2 minor concerns:

line 154: I would add a brief explanation of what transmission day or transmission hour means in the real world for the uneducated people

line 274: I think you mean to refer to figure 5.

A figure is not cited in the text, either figure 4 or 5 depending on the result of the previous comment.

Experimental design

no comment

Validity of the findings

no comment

Additional comments

I commend the authrs for the improvement on the readability of the text. It is now much more clear to read and understand.

I only suggest minor revision because a figure is never cited in the text but would not recommend further revision.

Reviewer 2 ·

Basic reporting

The manuscript is clear and unambiguous. Overall the English language is clear throughout; though there are some formatting issues - paragraph presentations, extra spaces, mix of US and UK English style that need fixing. The article structure is good. The manuscript is self contained.

Experimental design

The experimental design is clearly explained and in sufficient detail, with god rigour.

Validity of the findings

The findings are valid, with clear supporting rationale.

All listed points have been addressed.

Additional comments

The authors have addressed the comments of the reviewers, with the manuscript being greatly improved.

·

Basic reporting

The authors addressed my concerns.

Experimental design

The authors addressed my concerns.

Validity of the findings

The authors addressed my concerns.

Additional comments

Thank you for addressing my edits and comments.

---

## Round 0.3 · accepted · Accept

Authors have made those minor modifications recommended by reviewers. In addition, authors have used professional services to check and correct their English and grammar. Now this manuscript is acceptable.

#